# Oestrogen Non-Genomic Signalling is Activated in Tamoxifen-Resistant Breast Cancer

**DOI:** 10.3390/ijms20112773

**Published:** 2019-06-05

**Authors:** Coralie Poulard, Julien Jacquemetton, Olivier Trédan, Pascale A. Cohen, Julie Vendrell, Sandra E. Ghayad, Isabelle Treilleux, Elisabetta Marangoni, Muriel Le Romancer

**Affiliations:** 1Université de Lyon, F-69000 Lyon, France; coralie.poulard@lyon.unicancer.fr (C.P.); julien.jacquemetton@lyon.unicancer.fr (J.J.); olivier.tredan@lyon.unicancer.fr (O.T.); pascale.cohen@univ-lyon1.fr (P.A.C.); j-vendrell@chu-montpellier.fr (J.V.); sandra.ghayad@ul.edu.lb (S.E.G.); isabelle.treilleux@lyon.unicancer.fr (I.T.); 2Inserm U1052, Centre de Recherche en Cancérologie de Lyon, F-69000 Lyon, France; 3CNRS UMR5286, Centre de Recherche en Cancérologie de Lyon, F-69000 Lyon, France; 4Centre Léon Bérard, Medical Oncology Department, F-69000 Lyon, France; 5Solid Tumor Laboratory, Department of Pathology and Oncobiology, CHU Montpellier, 34000 Montpellier, France; 6Department of Biology, Faculty of Science II, EDST, Lebanese University, Fanar 90656, Lebanon; 7Centre Léon Bérard, Pathology Department, F-69000 Lyon, France; 8Translational Research Department, Institut Curie, 75005 Paris, France; elisabetta.marangoni@curie.fr

**Keywords:** breast cancer, resistance to endocrine therapy, oestrogen non-genomic signalling, PDX, biomarker

## Abstract

Endocrine therapies targeting oestrogen signalling have significantly improved breast cancer management. However, their efficacy is limited by intrinsic and acquired resistance to treatment, which remains a major challenge for oestrogen receptor α (ERα)-positive tumours. Though many studies using in vitro models of endocrine resistance have identified putative actors of resistance, no consensus has been reached. We demonstrated previously that oestrogen non-genomic signalling, characterized by the formation of the ERα/Src/PI3K complex, is activated in aggressive breast cancers (BC). We wondered herein whether the activation of this pathway is also involved in resistance to endocrine therapies. We studied the interactions between ERα and Src or PI3K by proximity ligation assay (PLA) in in-vitro and in-vivo endocrine therapy-resistant breast cancer models. We reveal an increase in ERα/Src and ERα/PI3K interactions in patient-derived xenografts (PDXs) with acquired resistance to tamoxifen, as well as in tamoxifen-resistant MCF-7 cells compared to parental counterparts. Moreover, no interactions were observed in breast cancer cells resistant to other endocrine therapies. Finally, the use of a peptide inhibiting the ERα–Src interaction partially restored tamoxifen sensitivity in resistant cells, suggesting that such components could constitute promising targets to circumvent resistance to tamoxifen in BC.

## 1. Introduction

Breast cancer (BC) is the most common cancer among women worldwide [1]. More than 75% of breast tumours express the oestrogen receptor α (ERα) in the nucleus and are commonly part of the group of luminal BCs. ERα plays a major role in BC tumorigenesis as it regulates cell cycle, cell survival and angiogenesis [2]. Interfering with the ERα pathway using anti-oestrogens (selective oestrogen receptor modulators such as tamoxifen or selective oestrogen down-regulators such as fulvestrant) or oestrogen deprivation (e.g., aromatase inhibitors), increases the survival of ERα-positive BC patients. Despite the high level of sensitivity of luminal tumours to endocrine therapy, treatment efficacy is limited by intrinsic and acquired resistance [3,4]. Indeed, 30–50% of patients relapse in the adjuvant setting and eventually die from metastases [5]. The main mechanisms underlying intrinsic resistance to tamoxifen are the lack of ERα expression and failure to convert tamoxifen into its active metabolite, while acquired resistance has been associated with a plethora of mechanisms. Those include alterations of ERα signalling, crosstalk between ERα and the growth factor network, activation of the PI3K–Akt–mTOR pathway, aberrant expression of cell-cycle regulators and induction of NFκB signalling [3,4]. Resistance to tamoxifen can also be due to the high expression of ERα-36, a splice variant of ERα expressed mainly at the plasma membrane of tumoral cells [6,7]. Indeed, ERα-positive tumours expressing high levels of ERα-36 appear to benefit less from tamoxifen treatment [8,9], probably because tamoxifen enhances stemness and promotes metastases via this receptor [10]. In addition, the G-protein coupled oestrogen receptor (GPER), an alternate oestrogen receptor with a different structure, has been associated with the development of tamoxifen resistance via a crosstalk with EGFR [11].

Oestrogen impacts normal and malignant breast tissue development via ERα, either through ligand-activated transcriptional regulation (genomic pathway) or by triggering cytoplasmic-signalling cascades (non-genomic pathway). Indeed, accumulating evidence indicates that a pool of ERα located in the cytoplasm and/or at the plasma membrane forms multiprotein complexes leading to the activation of downstream signalling molecules, such as MAPK and Akt [12,13]. Although several partners of extranuclear ERα have been described in different cell types, the most conserved partners are PI3K and tyrosine kinase Src [14,15,16]. Several years ago, our team reported that methylation of ERα on residue R260 by the arginine methyltransferase PRMT1 is a prerequisite for its association with Src and PI3K and the activation of Akt [17,18]. Subsequently, using the proximity ligation assay (PLA) methodology to detect in situ protein/protein interactions [19], we showed that this pathway, characterized by the formation of ERα/Src/PI3K and the subsequent activation of Akt, is present in normal breast tissue and is hyperactivated in aggressive breast tumours. This ERα/Src/PI3K multiprotein complex could be a new target for breast cancer treatments [20,21].

Current data regarding endocrine resistance has mostly arisen from BC cell lines able to adapt upon exposure to anti-oestrogens or aromatase inhibitors. However, such in-vitro models may not accurately reflect the complexity of interactions between BC cells and their microenvironment. The development of patient-derived xenograft (PDX) models of primary BC faithfully recapitulating the morphological and biological features of the parental tumours has revolutionised research in BC tumorigenesis, as evidenced by their extensive use over the last few years [22,23,24]. Recently, the team of E. Marangoni established two ERα-positive PDX models with acquired resistance to tamoxifen, constituting useful tools to identify markers of resistance [25].

In this study, we used both in vitro and in vivo models of endocrine resistance to investigate whether oestrogenic non-genomic signalling could be involved in resistance to endocrine therapies. We found that in both models the expression of ERα/Src/PI3K increases significantly and that targeting this complex partially restores tamoxifen sensitivity, opening new perspectives in the treatment of BC.

## 2. Results

### 2.1. ERα–Src–PI3K Expression in Patient-Derived Breast Cancer with Acquired Resistance to Tamoxifen

Since the ERα/Src/PI3K complex is highly expressed in aggressive ERα-positive breast cancers [20] and activation of the PI3K/Akt pathway is involved in resistance to endocrine therapies [26,27], we speculated that the activation of oestrogen non-genomic signalling was implicated in such resistance. Based on two tamoxifen-resistant PDXs, namely HBCx22 TamR and HBCx34 TamR, developed by the lab of Dr E. Marangoni from two primary hormone sensitive ERα-positive breast cancers, HBCx22 and HBCx34 [25], we assessed oestrogen non-genomic pathway activation by measuring the levels of interaction of ERα/Src and ERα/PI3K. Bright field PLA highlighted a significant increase in protein–protein interactions (as evidenced by brown dots) in the HBCx22 TamR model compared to parental PDXs (Figure 1A,B). Conversely, a non-significant decrease of ERα/Src/PI3K complex expression was recorded between the HBCx34 TamR compared to the parental PDX (Figure 1C,D), thus justifying the exclusive use of the HBCx22 TamR model for our subsequent experiments. In an attempt to understand the mechanism underlying the differences in these two resistant models, we analysed Src and PI3K expression in the different PDX models, and found that Src expression exclusively is increased in the HBCx22 TamR model compared to parental PDXs whereas no difference was observed in the HBC34 R model (Appendix A). In addition, as the ERα splice variant ERα-36 has also been linked with resistance to tamoxifen, we assessed its expression in the two models of resistance, but found that ERα-36 is more expressed in both resistant PDX models (Appendix A).

### 2.2. ERα/Src/PI3K Activation in HBCx22 TamR Treated with Everolimus and/or with Endocrine Therapies

The use of the mTOR inhibitor everolimus in combination with endocrine therapies in ERα-positive patients has significantly improved patient outcome [28,29]. The HBCx22 TamR PDXs were previously shown by Dr. E. Marangoni’s team to display a dual resistance to tamoxifen and fulvestrant, while the administration of everolimus alone or in combination with endocrine therapies to HBCx22 TamR mice resulted in a striking tumour growth inhibition of 90% [25]. Here, using tumour sections from treated xenografts, we analysed ERα and either Src (Figure 2A) or PI3K (Figure 2B). Quantification of PLA results clearly highlighted similar patterns of protein–protein interaction (Figure 2C), with tamoxifen and everolimus alone having little impact on these interactions, while their combination diminished both interactions. Furthermore, fulvestrant alone was as efficient as this latter combination, and the addition of everolimus did not enhance this effect (Figure 2C). However, Src and PI3K expression are not modified by the different treatments (Appendix A). These findings indicate that non-genomic signaling is associated with resistance to tamoxifen (but not to fulvestrant) in our HBCx22 TamR model.

### 2.3. ERα–Src–PI3K Interaction in MCF-7 Cells Resistant to Endocrine Therapy

Next, we wanted to assess whether we could confirm the activation of oestrogen non-genomic pathways in cellular models of resistance to endocrine therapies. We started our investigation using a model of MCF-7 cells resistant to tamoxifen (Res-Tam) that were established by exposing the control MCF-7 cells for 25 weeks to increasing concentrations of OH-Tam. We first verified the acquired resistance to tamoxifen by performing dose-dependent growth assays on cells depleted of steroids in charcoal-stripped serum, in order to precisely control the amounts of steroids in the medium (Figure 3A). When both MCF-7 control and Res-Tam cells were treated with increasing concentrations of tamoxifen from 10 ^−8^ M to 10 ^−5^ M, the survival rate was significantly higher for Res-Tam, as evidenced by the final values of 55% to 25% survival for Res-Tam and MCF-7 counterparts, respectively (Figure 3A). Interestingly, when we evaluated the interactions between ERα and Src or PI3K by PLA in both cell lines as previously described [20], those were more numerous in the tamoxifen-resistant cell line (Figure 3B,C). This increase in interaction is not due to a change in ERα expression or its translocation to the cytoplasm (Appendix A) nor to an increase of Src and PI3K expression (Appendix A). In addition, the downstream effector Akt is not activated in the MCF-7 resistant cell line compared to the parental cells (Appendix A).

We then performed similar experiments in MCF-7 cells resistant to fulvestrant (Res-Fulv), anastrozole (Res-Ana) and letrozole (Res-Let) (Figure 4). Compare to parental MCF-7 cells, Res-Tam cells alone displayed a significant increase in ERα/Src (Figure 4A,B) and ERα/PI3K interactions (Figure 4C,D).

### 2.4. Targeting the ERα/Src/PI3K Complex

Based on our results, we speculated that the oestrogen non-genomic complex played a role in cellular resistance to tamoxifen. To test this hypothesis, we decided to target the ERα/Src/PI3K complex using a peptide disrupting the ERα/Src interaction [30]. Indeed, the group of Aurricchio designed a six-amino peptide (pYpep) that mimics the sequence around the phosphotyrosine residue 537 of ERα impeding the ERα/Src interaction and oestrogen-induced cell proliferation [30]. We confirmed this finding and revealed that the use of the pYpep in MCF-7 cells completely disrupted the formation of the complex containing ERα/Src/PI3K [20]. Performing dose-dependent growth assays, we found that treating parental cells with the pYpep had no effect, while the peptide partially restored tamoxifen sensitivity in resistant cells (Figure 5A). This difference could be due to the milder effect of the pYpep on ERα/Src and ERα/PI3K interaction in the parental cells compared to resistant cells (Figure 5B).

## 3. Discussion

Our current study was triggered by the need to gain further insight into resistance to endocrine therapies in ERα-positive breast cancer, as despite ongoing improvements in patient survival and management, endocrine treatment efficacy is limited by intrinsic and acquired resistance. To circumvent such resistance, hormonotherapy must be combined with other treatments, the identification of which resides in a better understanding of the pathways implicated. In this study, using in cellulo and in vivo models, we found that non-genomic ERα signalling is upregulated synergistically to resistance to tamoxifen.

Indeed, components of this signalling pathway, namely mERα, Src and PI3K, previously shown to be implicated in BC [18], displayed a stronger activation, as evidenced in proximity ligation assays by their increased interactions [20] in a model of resistance to tamoxifen. Indeed, two PDX BC models resistant to tamoxifen were initially used to mimic in vivo conditions, namely HBCx-22 TamR and HBCx-34 TamR, and while activation of the oestrogen non-genomic pathway was clearly visible in the former model, we speculate that other molecular mechanisms underlie this resistance in the latter model. In addition, these models differed since the HBCx-22 TamR model displayed cross resistance to other endocrine treatments (fulvestrant and ovariectomy + letrozole), while the HBCx-34 TamR conserved a sensitivity to fulvestrant and partial response to ovariectomy and letrozole (19), suggesting different degrees of ERα inactivation in these models. Of note, in the HBCx22 Tam R model, activation of the non-genomic oestrogen signalling pathway did not impact phosphorylation of Akt [25], suggesting that other downstream signalling cascades may be involved. Analysis of transcriptomic profiles of HBCx22 TamR and HBCx34 TamR highlighted the activation of different transcription factors and transcriptional programs compared to the parental tumours. Based on our results, it appears that a subset of luminal tumours resistant to tamoxifen exhibit a dysregulation in oestrogen non-genomic signalling. Src could be a potential candidate, however, it would be necessary to extend the study to other models to identify predictive markers of the activation of the non-genomic pathway. However, we found that ERα-36 expression is increased in both PDX models of resistance comforting its role in resistance to tamoxifen [8,9].

Recent clinical studies showed that everolimus, the mTOR inhibitor, is beneficial to patients displaying resistance to endocrine therapy [29,31]. We showed that when treating the PDX models of endocrine resistance, the inhibition of tumour growth was more significant when combining endocrine therapy and everolimus compared to everolimus alone [25,31]. When we investigated ERα/Src and ERα/PI3K interactions on the fixed tumours from the different treatments, we found that fulvestrant triggers a strong decrease in both interactions, consistent with the fulvestrant-induced ERα degradation. The effect is milder with tamoxifen treatment probably due to a slight decrease in ERα expression assessed by IHC staining [25,31]. Conversely, everolimus had no effect, which was expected since its target is downstream of the non-genomic complex. Interestingly, both interactions decreased when tamoxifen was combined with everolimus, whereas ERα levels remained unchanged upon treatment with tamoxifen alone or in combination.

We then confirmed our results in MCF-7 cells resistant to tamoxifen. Surprisingly, activation of the oestrogen non-genomic pathway was specific to tamoxifen-resistant cells. Since neither the levels of ERα nor its cytoplasmic localization [32] differed in the resistant cell line, we hypothesize that tamoxifen may interfere with the formation of the complex, likely by modifying the enzymatic activity of PRMT1 or JMJD6, the arginine demethylase regulating the dynamics of ERα methylation [33].

Finally, the use of a competitive peptide partially restored tamoxifen sensitivity in the resistant cell line (Figure 5C), arguing in favour of targeting the oestrogen non-genomic complex in the case of ERα-positive breast tumors. However, the effect is not complete, suggesting that it could be improved either by inhibiting the activities of Src or PI3K shown to be important for the stabilization of the complex [34] and the level of ERα methylation [18], or by targeting PRMT1 activity, which we have recently shown to be involved in both ERα and IGF1-R signalling [35].

In conclusion, our work clearly demonstrates that (i) the level of activation of the non-genomic oestrogen ERα–Src–PI3K complex increases in different models of resistance to tamoxifen, and (ii) combining tamoxifen with Src or PI3K kinase inhibitors may improve patient survival by targeting both genomic and non-genomic oestrogen pathways.

## 4. Materials and Methods

### 4.1. Establishment of Xenografts Models Resistant to Tamoxifen (TamR)

HBCx22 and HBCx34 PDX models were established in female Swiss nude mice from untreated early-stage luminal breast cancers as previously described [25]. Both tumour models responded to tamoxifen treatment [36]. Luminal B status was confirmed both in patient tumours and derived xenografts, as evidenced by a low level of PR/high ki67 expression [36]. To generate hormone-resistant models from these xenografts, tumour-bearing mice were treated for 6 to 8 months with tamoxifen. Xenografts were deemed TamR once tumours had successfully been engrafted in three consecutive mice (serial passages) and exhibited a resistant phenotype [25]. Tissue microarrays (TMA) were then constituted by including parental and TamR xenografts as well as in-vivo endocrine therapy-treated tumours as previously described below. Three xenografts from each treatment group and two tissue cores per tumour were included in the TMA.

### 4.2. In Vivo Treatment Studies

Treatment studies were performed in accordance with the French Ethics Committee as detailed in Cottu et al. 2014 (02163.01, November 2014) [25]. HBCx22 TamR were either administered tamoxifen (4 mg/kg/week); fulvestrant (50 mg/kg/week); or everolimus (Novartis Pharma), an mTOR inhibitor, orally at a dose of 2.5 mg/kg three times/week, over 100 days. Tumours harvested at the end of treatments were embedded in TMA, as described above.

### 4.3. Cell Culture

From the ERα MCF-7-derived breast cancer cell line stably transfected with the human aromatase gene (MCF-7 control) as previously reported [37], different cell lines were established, namely: a tamoxifen-resistant cell line (Res-Tam), a fulvestrant-resistant cell line (Res-Fulv), an anastrozole-resistant cell line (Res-Ana) [38], and a letrozole-resistant cell line (Res-Let) [39]. The Res-Tam cell line was established by exposing the MCF-7 control cells during 25 weeks to increasing concentrations (1 and 3 μM) of 4-hydroxy-tamoxifen (OH-Tam, Sigma, St Louis, MO, USA) in Dulbecco’s Modified Eagle Medium without phenol red, supplemented with 3% steroid depleted, dextran-coated and charcoal-treated fetal calf serum (DCC medium) containing 25 nM 4-androstenedione (AD) (Sigma, Saint Louis, MO, USA). All of the human cell lines were maintained at 37 °C in the appropriate medium supplemented with 10% foetal calf serum.

### 4.4. Antibodies 

The details of primary antibodies used in this study are listed in the Table 1.

**Table 1 ijms-20-02773-t001:** Information of primary antibodies.

Antibodies	Supplier	Origin	Dilution for PLA	Dilution for WB	Dilution for IF	Dilution for IHC
PI3K p85 ab-22653	Abcam	mouse	1/30			
c-Src (B12) sc-8056	Santa Cruz	mouse	1/150	1/1000		1/1000
ERα (HC20) sc-542	Santa Cruz	rabbit	1/75			
ERα (F10) sc-542	Santa Cruz	mouse		1/200	1/200	
GAPDH Sc-47724	Santa Cruz	mouse		1/2000		
PI3K p85 05-212	Millipore	mouse		1/1000		1/100
ERα-36	Custom made [40]	rabbit				1/100
p-Akt 4060	Cell Signaling	rabbit		1/1000		
Akt 4691	Cell Signaling	rabbit		1/1000		

PLA: Proximity Ligation Assay. IF: Immunofluorescence. WB: Western Blotting. IHC: immunohistochemistry.

### 4.5. Proximity Ligation Assay

This technology, first published in 2006 [19], enables the in-situ visualisation of protein interactions and was supplied by Merck.

For fluorescent microscopy, cells were grown on coverslips in 12-well plates and fixed in methanol. Cells were saturated using the blocking solution, then antibodies against ERα, Src or PI3K were incubated for 1 h at 37 °C. After washes, the PLA probes were incubated for 1 h at 37 °C. Following the ligation of the oligonucleotides, an amplification reaction was performed for 100 min at 37 °C. The samples were then mounted with Duolink II Mounting Medium containing 4′6′-diamidino-2-phenylindole (DAPI) and then visualised under fluorescent microscopy.

For bright field microscopy, paraffin-fixed tumour tissues were incubated in a hydrogen peroxide solution, for 5 min at room temperature, to avoid peroxidase quenching. For antibody detection, the probes were labelled with horseradish peroxidase after two washes in high purity water. Samples were mounted in non-aqueous mounting medium and visualised under a bright field microscope.

### 4.6. Western Blotting

Cells were lysed using RIPA buffer (50 mM Tris HCl, pH 8, 150 mM NaCl, 1 mM EDTA, 1% NP-40 and 0.25% deoxycholate) supplemented with protease-inhibitor tablets (Roche Molecular Biochemicals, Basel, Switzerland) and phosphatase inhibitors (1 mM NaF, 1 mM Na_3_VO_4_ and 1 mM b-glycerophosphate). Cell extracts were separated on SDS-PAGE and visualised by ECL.

### 4.7. Immunofluorescence

MCF-7 cells (7 × 10^4^) were grown on coverslips in 12-well plates. After treatment, cells were fixed in methanol for 2 min and washed twice in PBS. Non-specific binding was blocked using a 1% gelatin solution for 30 min at room temperature and cells were incubated with the anti-ERα antibody for 1 hr at 37 °C, then with the secondary antibody Alexa Fluor 488 (Molecular Probes, Carlsbad, CA, USA) in Dako diluent, and mounted on glass slides in mounting solution (Dako, Carpinteria, CA, USA). The images were acquired using a fluorescent microscope.

### 4.8. Cytotoxicity Assay

Cells were grown for 4 days in a hormone-stripped DMEM devoid of phenol red and supplemented with 3% steroid-depleted, charcoal-treated foetal calf serum (Biowest, Riverside, MO, USA); 10^4^ cells per well were plated onto a 96-well plate in charcoal-stripped medium. The following day, the cells were treated with 4-hydroxy-tamoxifen (OH-Tam, Sigma) at different concentrations for 72 h. Cell viability was analysed using the Cell Titer 96^®^ aqueous one solution Cell Proliferation kit from (Promega, Fitchburg, WI, USA). Briefly, the reagent (MTS) was added in each well, incubated for 1 h at 37 °C, and the absorbance was recorded at 490 nm following the manufacturer’s recommendations.

### 4.9. Image Acquisition and Analysis

The hybridized fluorescent slides were viewed under a NIKON NIE microscope. Images were acquired under identical conditions at X60 magnification with different filters (DAPI, GFPHQ and Cy3). On each sample, 100 cells were counted. Analyses and quantifications of these samples were performed using the Image J software (Version 1.52, NIH, Bethesda, MD, USA). This software includes a “Counter cells” plugin, which enables users to count and analyse the number of cells/dots present on an 8-bit color image.

The hybridized bright field slides were viewed under a Leica DMLB microscope. Images of three independent zones on each tumour TMA section were acquired under identical conditions at X40 magnification. At least 500 cells were counted per tumour.

## Figures and Tables

**Figure 1 ijms-20-02773-f001:**
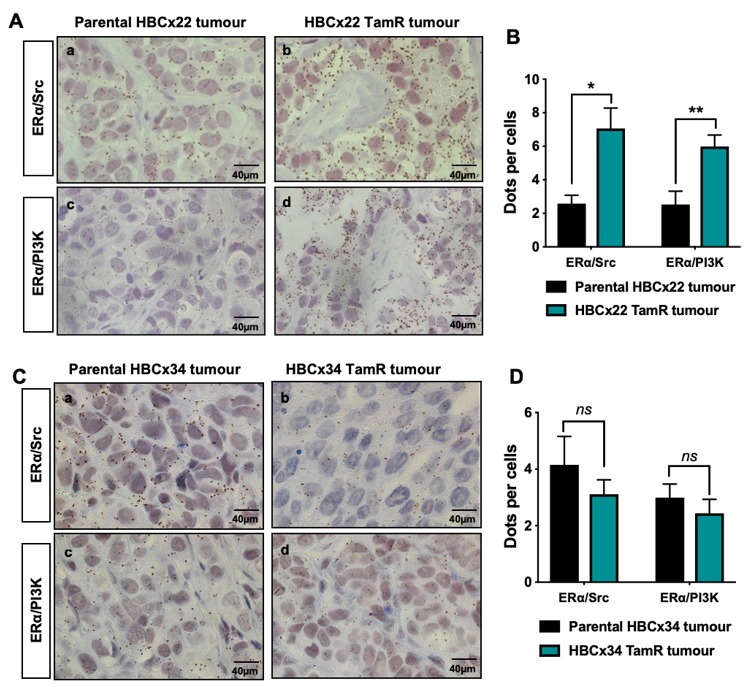
ERα/Src and ERα/PI3K interactions in tamoxifen-resistant breast cancer PDX models. Bright field proximity ligation assay (PLA) was performed on tumours embedded in tissue microarrays (TMA) to study the interactions between ERα and Src or PI3K in the parental using the antibodies listed in Table 1 (**A**) HBCx22 or (**C**) HBCx34 tumours compared to the tamoxifen-resistant (TamR) tumours. The brown dots represent protein–protein interactions (x 40 magnification). (**B,D**) Quantification of results presented in (**A**,**C**) was performed by counting the number of signals per cell in three independent zones of the section (*n* ≥ 500 cells counted/tumour). Significance (*p*-value) between treatments was determined using the Student *t*-test. ns: non-significant; * *p* < 0.05; ** *p* < 0.01.

**Figure 2 ijms-20-02773-f002:**
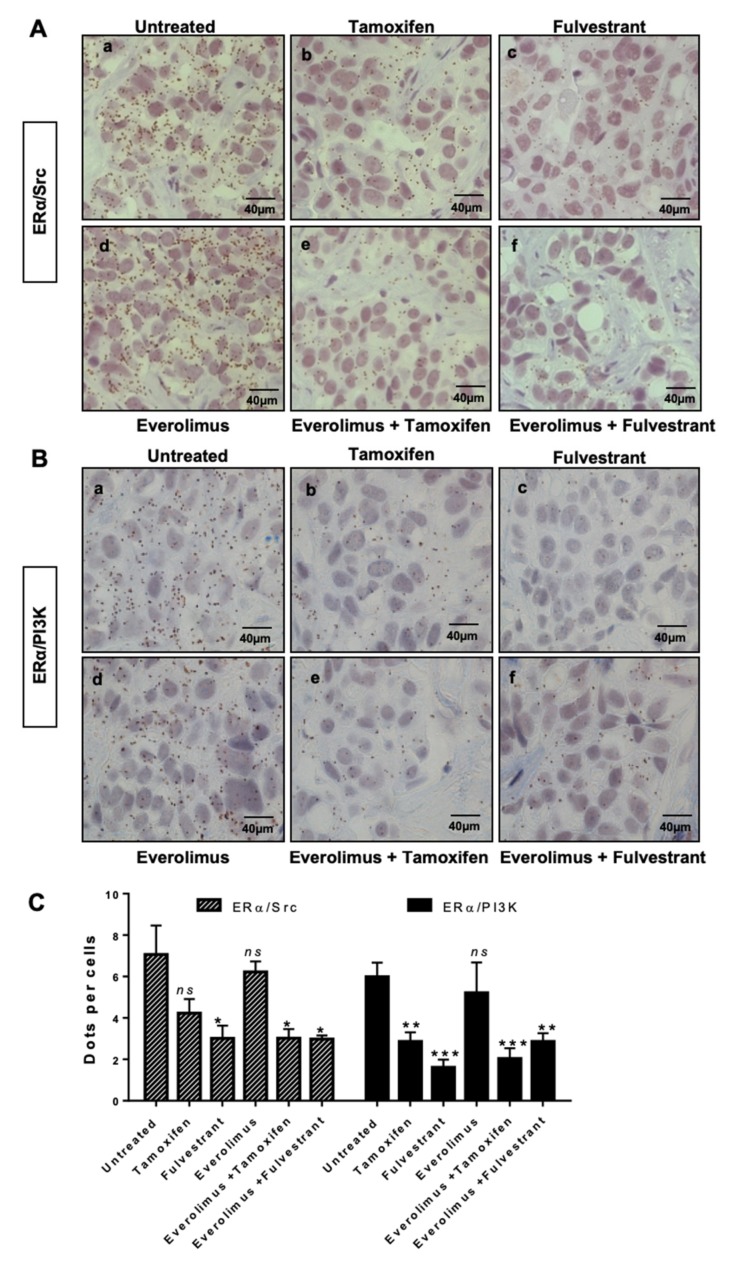
ERα/Src and ERα/PI3K interactions in HBCx22 TamR PDX model treated with everolimus combined or not with endocrine therapies. HBCx22 TamR mice were treated with tamoxifen, fulvestrant, everolimus, everolimus, and tamoxifen or everolimus, and fulvestrant prior to being sacrificed. The tumours were then embedded in paraffin and included in tissue micro arrays (TMA). Bright field proximity ligation assays were subsequently conducted on TMA sections to analyse the interactions between (**A**) ERα and Src or between (**B**) ERα and PI3K. n=3 xenografts/group (**C**). Quantification of results obtained was performed by counting the number of signals per cell in three independent zones of the section. (*n* ≥ 500 cells counted/tumour). Significance (*p*-value) between treatments was determined using the Student *t*-test. *** *p* < 0.001.

**Figure 3 ijms-20-02773-f003:**
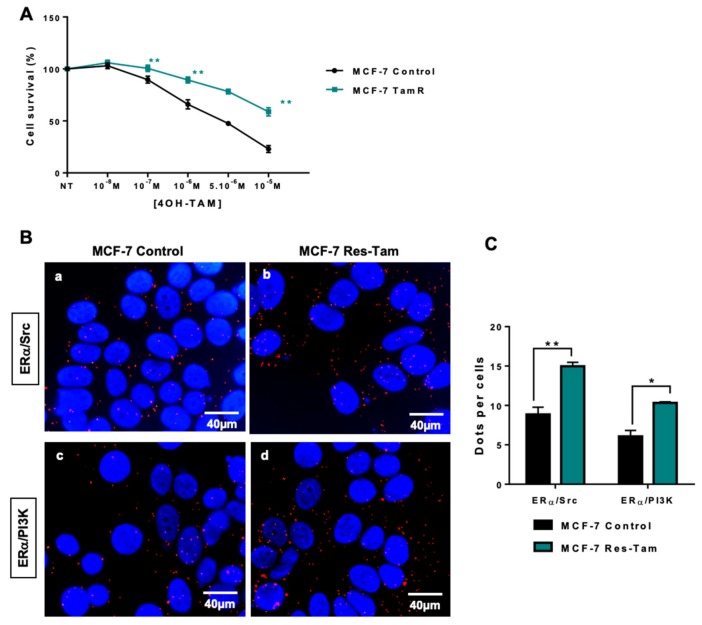
ERα/Src and ERα/PI3K interactions in tamoxifen-resistant MCF-7 cells. (**A**) Tamoxifen sensitivity was studied in parental and tamoxifen-resistant MCF-7 cell lines (MCF-7 Res-Tam) grown in hormone-depleted conditions and treated with increasing doses of tamoxifen prior to being analysed using the MTS assay. (**B**) After fixation of cells in A, proximity ligation assays were performed to evaluate ERα/Src and ERα/PI3K interaction in both cell lines. The detected dimers are represented by red dots, while the nuclei were counterstained with DAPI (blue) (x63 magnification). (**c**) Quantification of results obtained in A and B was performed by counting the number of signals per cell (n = 100 cells) using the image J “counter cells” plugin. The graph is representative of three independent experiments. Significance (*p*-value) between treatments was determined using the Student t-test. * *p* < 0.05; ** *p* < 0.01.

**Figure 4 ijms-20-02773-f004:**
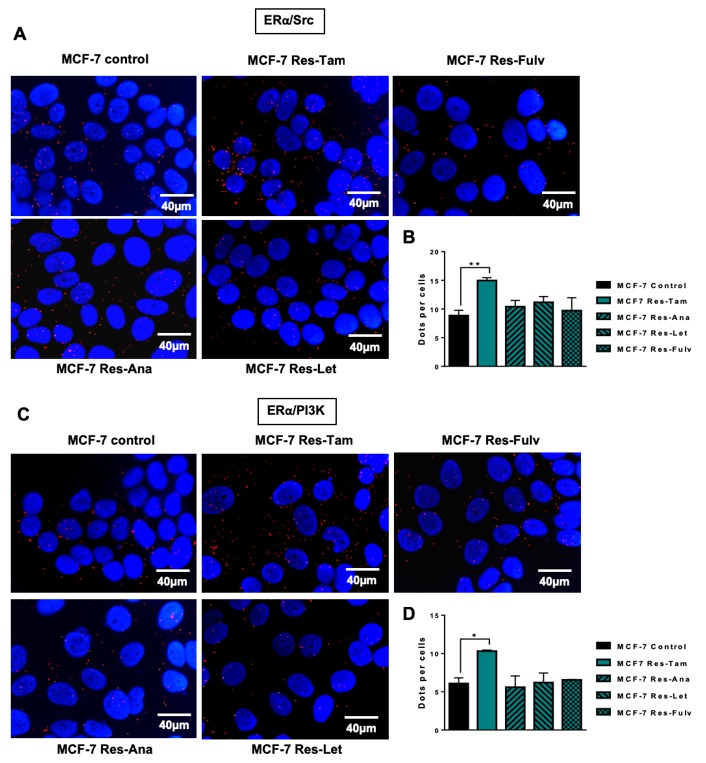
ERα/Src and ERα/PI3K expression in MCF-7 cells resistant to several endocrine therapies. MCF-7 cells resistant to tamoxifen (Res-Tam), fulvestrant (Res-Fulv), anastrozole (Res-Ana) and letrozole (Res-Let) were generated and the levels of interaction between (**A**) ERα/Src or (**C**) ERα/PI3K were determined by proximity ligation assay using fluorescent probes (**B**,**C**). Quantification of results obtained in (**A**,**C**) was performed by counting the number of signals per cell (*n* = 100 cells) using the image J “counter cells” plugin. The graph is representative of three independent experiments. Significance (*p*-value) between treatments was determined using the Student t-test. * *p* < 0.05; ** *p* < 0.01.

**Figure 5 ijms-20-02773-f005:**
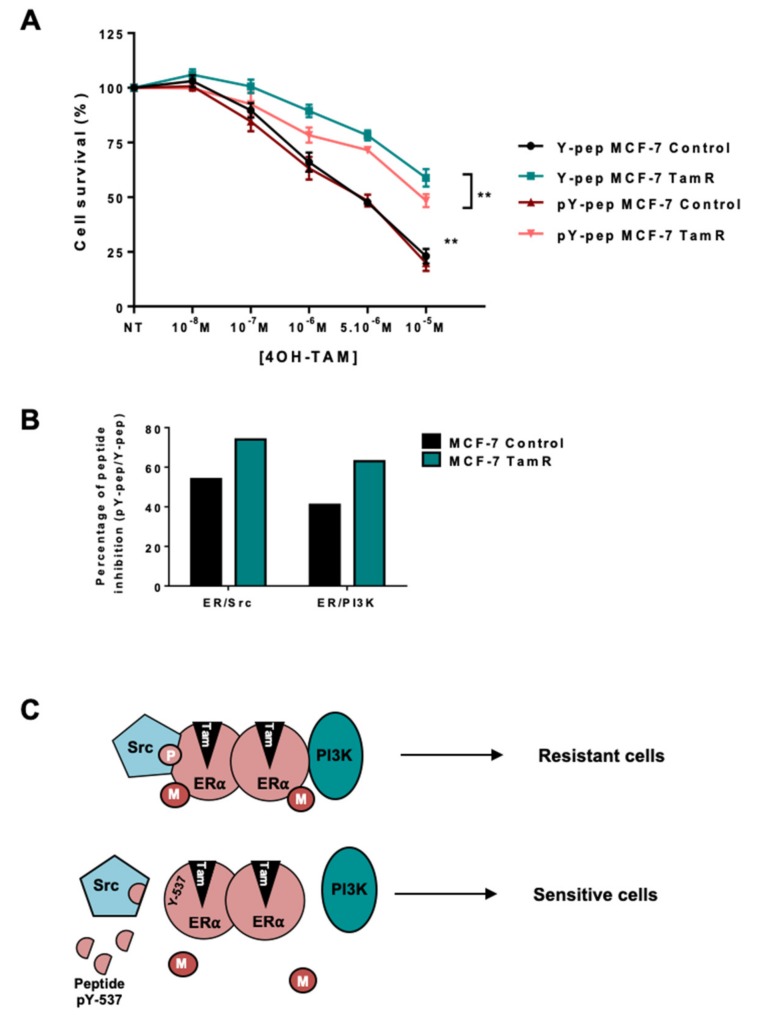
Targeting the ERα/Src dimer in MCF-7 Res-Tam cells. (**A**) Parental and Res-Tam MCF-7 cells were incubated with 1 nM of a peptide mimicking hERα 536-541 containing Y537 (Y-pep) or the corresponding phosphorylated peptide (pY-pep) for 30 min prior to treatment. We then assessed their response to increasing concentrations of tamoxifen using an MTS cytotoxic assay. (**B**) ERα/Src and ERα/PI3K interactions were determined by proximity ligation assay after treating MCF-7 cells with pY-pep or Y-pep. Quantification of the number of dots per cells was performed as previously described and the percentage of peptide inhibition was calculated (pY-pep/Y-pep). (**C**) Model of oestrogen non-genomic signalling in tamoxifen-resistant BC. M corresponds to methylation and P to phosphorylation.

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
