# Peer review of "Oestrogen Non-Genomic Signalling is Activated in Tamoxifen-Resistant Breast Cancer"

_ijms, 2019, doi:10.3390/ijms20112773_

Round 1

Reviewer 1 Report

The authors have submitted a revised version of their manuscript, corrected several points and added additional data.

I think the manuscript has been improved greatly, and only a few minor points remain:

I cannot follow the remark on Ignatov´s 2019 paper, which clearly states “GPER-expression is associated with poor response to tamoxifen”.  I agree that the function of GPER1 for tam-resistance is debated, but I would also extend that opinion to ERalpha36.

There is one reference, giving actual data on tamoxifen in tissue under treatment:

Kisanga ER, Gjerde J, Guerrieri-Gonzaga A, et al. Tamoxifen and metabolite concentrations in serum and breast cancer tissue during three dose regimens in a randomized preoperative trial. Clin Cancer Res Off J Am Assoc Cancer Res. 2004;10:2336–2343.

I also think, one should cite the Itoh et al 2005 paper (Mol Cancer Res 2005;3(4).) or Sun et al. J. Steroid Biochem. Molec. Biol. Vol. 63, No. 1-3, pp. 29-36, 1997 in relation to the MCF-7aro cell line.

I am not sure what a “home-made” antibody is. It has been obtained from somewhere or was freshly generated by immunization with the usual peptide?!

Author Response

Response to reviewer 1:

-We did not speak about GPER because we are not convinced that it acts as an estrogen receptor.

- We added the reference for the MCF-7 aro cell line.

- We modified “home made” to “custom made”

Reviewer 2 Report

All remarks taken into account. Paper amended adequately

Remarks of minor importance  for final writing .

L101-103: Src and PI3K expression in HBCx22 TamR model : text refers solely to an increase of   Src in  support of  a corresponding visual impression   ( Fig S1) . If this would not be so ( at least  an increase of lower extent for PIK3) state for  the usual Src/PI3K which a notification.

L102 : remove " observe" . Corrrect sentence :  ..no difference was observed...

Author Response

Response to reviewer 2:

We have modified the text as requested.

This manuscript is a resubmission of an earlier submission. The following is a list of the peer review reports and author responses from that submission.

Round 1

Reviewer 1 Report

ERa-induced signaling cascades of a pool of extra nuclear receptors ( mainly localized at the plasma membrane) is recognized as a potent process  promoting hormone dependence of breast cancer. Hence, alteration at his level  may logically generate resistance to endocrine therapy, especially anti-estrogen administration. Present paper provide original data supporting this concept. Reported experimental approaches (use of patients-derived  xenografs) as well as specifically investigated molecular mechanism ( implication of ERa R260 methylation in Src/PI3K) not usually adressed, justify its report.

Underlying remarks should nevertheless, be taken into account,

#2.1 L89 .    " ERa/Src/PI3K complex activation  was not significantly different between HBCx34 and HBCTx34 Tam R". This sentence needs to be modified since a non significant  decrease is recorded between parental and Tam R  instead of an evident (expected) increase in the HBCx22 model.  A distinct ( complemetary ) antagonistic procedure may be at the origin of this  behavior  ( see remark below). Absence of any data concerning the relevance of this hypothesis justifies the sole assessment of the HBCx22 model.

#2.2  Text L98-105 needs to be improved. At first, quantification of PLA data under Tam  treatment  (3C)  fails to coborate  corresponding visual records(3A) : unsignificant decrease  vs.  lack of effect. Morever, Text should clearly state that  combinaison of Everolimus and antiestrogen  is solely effective when the effect of the atiestrogen is weak ( Tam).

#2.3/4   MCF-7 Tam Res cell line . Provide  origin/ propetries .. Alparently not recorded in ref 31  and 32 ( sole accessible data in Suppl Fig)

#2.4  p Y-pep . Study in ref 14 reveal a strong antagonism  of this peptide  on the ERa/SRc/PI3K interaction. In contrast, data of Fig 5A suggest solely a mild restoration of Tam antagonism.  This need some comments  concerning  the potential valididity of this therapeutic approach !( discussion). Morover,  could a complementary assessment of a cumulative effect of  Everolimus be incorporated in the present investigation?

Complementary remarks

ERa most likely refers to the 67KDa  wild type receptor ( not specified on Fig1S). A 36 KDa non nuclear   ERa  variant  identified in Tam resitant tumors is subjected to increasing interest ( review :Teymourzadeh Clin Breast Cancer17:403, 2017; several other papers)   In the investigated xenografs, the presence of such a variant  has most probably not been  been assessed ? Advocation for such an investigation ( discussion) would increase the reading   impact of the paper. It may also perhaps justify  the recorded difference between HBCx 22and 34.

#2.2  L99 " we analysed ERa and either Src..  ."  we assessed the interaction of ERa with Src..".   L104 " findings indicated  that non genomic signaling  appears to be..." :  indicate and appaers" do not fit well together !

Fig 5B ( summary model)  displace to the discussion ~L181

Author Response

Reviewer 1

ERa-induced signaling cascades of a pool of extra nuclear receptors ( mainly localized at the plasma membrane) is recognized as a potent process  promoting hormone dependence of breast cancer. Hence, alteration at his level  may logically generate resistance to endocrine therapy, especially anti-estrogen administration. Present paper provide original data supporting this concept. Reported experimental approaches (use of patients-derived  xenografs) as well as specifically investigated molecular mechanism ( implication of ERa R260 methylation in Src/PI3K) not usually adressed, justify its report.

Underlying remarks should nevertheless, be taken into account,

#2.1 L89 .    " ERa/Src/PI3K complex activation  was not significantly different between HBCx34 and HBCTx34 Tam R". This sentence needs to be modified since a non significant  decrease is recorded between parental and Tam R  instead of an evident (expected) increase in the HBCx22 model.  A distinct ( complemetary ) antagonistic procedure may be at the origin of this  behavior  ( see remark below). Absence of any data concerning the relevance of this hypothesis justifies the sole assessment of the HBCx22 model.

Response:

We have amended the text as requested.

#2.2  Text L98-105 needs to be improved. At first, quantification of PLA data under Tam  treatment  (3C)  fails to coborate  corresponding visual records(3A) : unsignificant decrease  vs.  lack of effect. Morever, Text should clearly state that  combinaison of Everolimus and antiestrogen  is solely effective when the effect of the atiestrogen is weak ( Tam).

Response:

We have changed Figure 2A (we believe the reviewer may have mistaken 3A with 2A) and have modified the text consequently.

#2.3/4   MCF-7 Tam Res cell line . Provide  origin/ propetries .. Alparently not recorded in ref 31  and 32 ( sole accessible data in Suppl Fig)

Response:

The cell line has been established by Dr Cohen, coauthor of this article, and has not been published before. The Res-Tam cell line was established by exposing the MCF-7aro cells during 25 weeks to increasing concentrations (1 and 3 μM) of 4-hydroxy-tamoxifen (OH-Tam, Sigma, St Louis, MO) in Dulbecco’s Modified Eagle Medium without phenol red, supplemented with 3% steroid depleted, dextran-coated and charcoal-treated fetal calf serum (DCC medium) containing 25 nM 4-androstenedione (AD) (Sigma). This has now been added to the Materials and Methods.

#2.4  p Y-pep . Study in ref 14 reveal a strong antagonism of this peptide  on the ERa/SRc/PI3K interaction. In contrast, data of Fig 5A suggest solely a mild restoration of Tam antagonism.  This need some comments  concerning  the potential valididity of this therapeutic approach !( discussion). Morover,  could a complementary assessment of a cumulative effect of  Everolimus be incorporated in the present investigation?

Response:

This result has been argued in the discussion section as requested.

We did not treat the cells with everolimus as it has no effect on ERα/Src and ERα/PI3K interactions in the PDX model.

Complementary remarks

ERa most likely refers to the 67KDa  wild type receptor ( not specified on Fig1S).

Response:

For us ERα refers to the full length ERα66 kDa

-A 36 KDa non nuclear   ERa  variant  identified in Tam resitant tumors is subjected to increasing interest ( review :Teymourzadeh Clin Breast Cancer17:403, 2017; several other papers)   In the investigated xenografs, the presence of such a variant  has most probably not been  been assessed ? Advocation for such an investigation ( discussion) would increase the reading   impact of the paper. It may also perhaps justify  the recorded difference between HBCx 22and 34.

Response:

We thank the author for this suggestion. We assessed ERα-36 expression by performing IHC experiments in the PDX models and found that its expression is increased in the 2  models comforting its involvement in resistance to tamoxifen (see Figure S1).

#2.2  L99 " we analysed ERa and either Src..  ."  we assessed the interaction of ERa with Src..".   L104 " findings indicated  that non genomic signaling  appears to be..." :  indicate and appaers" do not fit well together !

Response:

We have modified the text as requested.

Fig 5B ( summary model)  displace to the discussion ~L181

Response:

We would prefer to keep the figure as originally submitted.

Reviewer 2 Report

In this manuscript, the authors hypothesized that an interaction between ER-alpha, SRC and PI3K is involved in tamoxifen resistance. The authors investigated this interaction in endocrine resistant breast cancer models by proximity ligation assay and application of a peptide that is supposed to block this interaction.

The authors found that the ER-Src-PI3K interaction indeed occurred in tamoxifen resistant models (one PDX, one cell line) and that the blocking peptide has an impact on tamoxifen resistance.

This is an overall interesting paper, contributing to the ongoing discussion on the mechanism of acquired tamoxifen resistance. The manuscript fits well to the scope of the journal; however, I have some points that should be clarified before publication.

Introduction:

Surprisingly, GPER and ERa36 were not mentioned as a possible resistance mechanism.

Results:

From the two PDX models analyzed, only one showed the interaction in PLA to be increased after tamoxifen treatment, although both models started with comparable values. Overall expression levels of the interacting molecules are not given, but would be helpful for interpreting the results of the two PDX models.

This is also important for the abundance of the PLA-markers under treatment, such as fulvestrant as it has significant impact on ERalpha expression and localization.

MCF-7aro cells were adapted to tamoxifen and found to be “resistant”, however at concentrations above 100nM. Can such concentrations be reached in tumors under treatment?  Interestingly, the cells were also depleted of estrogen and related compounds by using charcoal stripped serum for this assay. Does it make sense to remove the proliferative signal that is supposed to be blocked by the inhibitor? How is this related to the use of the “MCF-7aro” cell line? Please explain this rational further, it is not enough to mention this in the method section.

These MCF-7aro TamR cells, however also showed an increase in the PLA-markers. Again, please show the overall protein expression, not only the ER.

It is also not clear whether the interaction indeed results in signaling activity, Can the authors provide further evidence for this?

The effect of the blocking peptide is quite intriguing. Already in the publication by Varricchio et al it was shown that the blocking peptides are taken up into the cells and are not hydrolyzed or dephosphorylated, which I still find quite surprising. Nevertheless, the peptides showed an impact on OH-Tam sensitivity, where both peptides had no effect in MCF-7 and the phosphorylated sensitized the MCF-7 Tam cell line to some extent. Please show how this treatment influences the PLA marker abundance.

Please include scale bars into the micro-photographs.

Methods:

It is not clear how long the cells were incubated with 4OH-TAM for sensitivity assays. In the methods it is OH-Tam and not tamoxifen, when this is true; please add this information also to the figure legend and the axis.

Image acquisition: I think it is generally recommended to describe filter sets used for fluorescence microscopy.

Author Response

Reviewer 2

Comments and Suggestions for Authors

In this manuscript, the authors hypothesized that an interaction between ER-alpha, SRC and PI3K is involved in tamoxifen resistance. The authors investigated this interaction in endocrine resistant breast cancer models by proximity ligation assay and application of a peptide that is supposed to block this interaction.

The authors found that the ER-Src-PI3K interaction indeed occurred in tamoxifen resistant models (one PDX, one cell line) and that the blocking peptide has an impact on tamoxifen resistance.

This is an overall interesting paper, contributing to the ongoing discussion on the mechanism of acquired tamoxifen resistance. The manuscript fits well to the scope of the journal; however, I have some points that should be clarified before publication.

-Introduction: Surprisingly, GPER and ERa36 were not mentioned as a possible resistance mechanism.

Response:

We thank the referee for this pertinent remark. We added information about the role of ERα-36 in the development of resistance to endocrine therapy in the introduction section. We also assessed ERα-36 expression as suggested by the reviewer 1 and found that its expression is increased in the 2 PDX resistant to tamoxifen (see Figure S1).

 However, we did not speak about GPER because its role in the development of tamoxifen resistance in breast cancer is controversial. As an example, a recent publication showed that GPER expression is a favorable prognostic factor. (Ignatov et al, 2019)

Results:

-From the two PDX models analyzed, only one showed the interaction in PLA to be increased after tamoxifen treatment, although both models started with comparable values. Overall expression levels of the interacting molecules are not given, but would be helpful for interpreting the results of the two PDX models.

Response:

The referee is right, it is an interesting point to investigate. Concerning ERα, we know that its expression is not modified in the Tam R PDX versus the parental models (From Cottu et al, CCR, 2014). For Src and PI3K expression, we performed IHC experiments with the corresponding antibodies and found that Src expression is increased in HBCx 22 Tam R compared to the parental PDX but not in the HBC34 R model, possibly explaining the difference observed in the 2 models of resistance.  These results have been shown in Figure S2.

This is also important for the abundance of the PLA-markers under treatment, such as fulvestrant as it has significant impact on ERalpha expression and localization.

Response:

ERα expression has been assessed by Dr Marangoni’s team in Cottu et al, 2014. Concerning the other PLA markers, we have investigated their expression by IHC. The results are shown in Figure S2 and do not present any difference.

-MCF-7aro cells were adapted to tamoxifen and found to be “resistant”, however at concentrations above 100nM. Can such concentrations be reached in tumors under treatment?  Interestingly, the cells were also depleted of estrogen and related compounds by using charcoal stripped serum for this assay. Does it make sense to remove the proliferative signal that is supposed to be blocked by the inhibitor? How is this related to the use of the “MCF-7aro” cell line? Please explain this rational further, it is not enough to mention this in the method section.

In vitro resistance is generally observed with 100 nM in different cell models of acquired resistance to OH-Tam (MTLN and its resistant clones R7, R8) [Badia E et al, Cancer Res, 2000] or Tam (VP229/VP267) [McCallum HM, Breast Cancer Res Treat, 1996]. There is no data indicating the amount of Tamoxifen that can be reached in tumours under treatment. However, we can provide the information that the Tam resistant cell line VP267, obtained after in vivo resistance to Tamoxifen, is resistant to 200 nM of OH-Tam in vitro [Ghayad SE et al, J Mol Endocrinol, 2009], suggesting that such concentrations could be reached in patients.

The MCF-7 cells were stably transfected with the human aromatase gene (MCF-7 Aro). Starting from here, to preserve the background, the cells were then selected for Tamoxifen-resistance or fulvestrant-resistance or anastrozole-resistance or letrozole-resistance. The cells were depleted for steroids in order to control the amount of steroids the cells are treated with. This information has been added to the manuscript. In addition to avoid any confusion, we called the parental MCF-7 used to establish the resistant cells (previous MCF7 Aro) : MCF-7 control.

-These MCF-7aro TamR cells, however also showed an increase in the PLA-markers. Again, please show the overall protein expression, not only the ER.

Response:

As requested, we analysed the level of expression of Src and PI3K in MCF7 Tam R and the parental cells. The result have been added in the Supplemental Figure 3.

- It is also not clear whether the interaction indeed results in signaling activity, Can the authors provide further evidence for this?

Response,

As requested, we have analysed Akt activation in both cell lines by performing Western blotting using anti-Akt and p-Akt antibodies. As shown in Supplemental Figure 3, P-Akt is not modified in Tam R cell versus the parental MCF-7 similarly to what was found in the PDX model of resistance to tamoxifen (Cottu et al 2014).

-The effect of the blocking peptide is quite intriguing. Already in the publication by Varricchio et al it was shown that the blocking peptides are taken up into the cells and are not hydrolyzed or dephosphorylated, which I still find quite surprising. Nevertheless, the peptides showed an impact on OH-Tam sensitivity, where both peptides had no effect in MCF-7 and the phosphorylated sensitized the MCF-7 Tam cell line to some extent. Please show how this treatment influences the PLA marker abundance.

Response:

We have studied the PLA marker abundance and shown the result in Fig 5B.

 -Please include scale bars into the micro-photographs.

Response:

The scale bars have been added as requested. 

-Methods: It is not clear how long the cells were incubated with 4OH-TAM for sensitivity assays. In the methods it is OH-Tam and not tamoxifen, when this is true; please add this information also to the figure legend and the axis.

Response:

We added the time of treatment in the method section and TAM was replaced by OH-TAM in the figure legend and axis.

-Image acquisition: I think it is generally recommended to describe filter sets used for fluorescence microscopy.

Response:

We have added this information in the method section.